# Slack Resources, Corporate Performance, and COVID-19 Pandemic: Evidence from China

**DOI:** 10.3390/ijerph192114354

**Published:** 2022-11-02

**Authors:** Ling Jin, Jun Hyeok Choi, Saerona Kim, Kwanghee Cho

**Affiliations:** 1Department of Accounting, Yanbian University, 977 Gongyuan Rd., Yanji 133002, China; 2Department of Accounting, Dongguk University, 30 Pildongro 1-gil, Jung-gu, Seoul 04620, Korea; 3Department of Tax & Accounting, Gyeongsang National University, 501 Jinju-daero, Jinju-si 52828, Korea

**Keywords:** COVID-19, unabsorbed slack, absorbed slack, potential slack, financial constraints, China

## Abstract

COVID-19 has caused tremendous damage to global economies, and similar health crises are expected to happen again. This study tests whether slack resources would enable companies to prepare for such uncertainties. Specifically, we explored the influence of the COVID-19 patient occurrence on corporate financial performance and the buffering effect of financial slacks using Chinese listed companies’ data during 2021. We also examined whether this effect differs across firms’ financial health and industry. Test results are as follows. First, consistent with the recent studies on pandemics, the degree of COVID-19 prevalence had a negative impact on the Chinese company’s financial performance, and slack resources offset this adverse effect. Second, slack’s buffering effects appeared mostly in financially constrained companies. Third, such effects mostly appeared in industries vulnerable to the COVID-19 shock. In the business environment of 2021, adapted to COVID-19, our main test result seems to mainly come from companies with a greater need for slack. Our tests imply that, despite differences in the degree of accessibility to resources, excess resources help companies overcome the COVID-19 crisis, which means that firms can more efficiently respond to economic shocks such as COVID-19 if they reserve past profits as free resources. This study contributes to the literature in that there is limited research on the slack resources’ buffering effect on the COVID-19 shock and that this study works as a robustness test as it uses data from one of the East Asian regions at a time when the control of COVID-19 was relatively consistent and successful, which can limit the effect of COVID-19 and slacks.

## 1. Introduction

On 11 March 2020, when the World Health Organization declared the outbreak of the coronavirus disease 2019 (COVID-19) pandemic, the global economy experienced heavy damage [1,2]. COVID-19 had a more substantial and widespread impact on companies than any previous epidemics [1]. Major countries such as the United States, United Kingdom, Germany, France, and Japan suffered a 2 to 5% decline in Gross Domestic Product (GDP) [3]. China suffered a more profound shock during this period, with a 6.1% decline in GDP compared with the same period in the previous year [4].

The start of the COVID-19 shock in Chinese companies was earlier than in other countries, at least from the Wuhan lockdown on January 23. According to Gu et al. [3]’s study of electricity consumption in Suzhou, located near Shanghai, electricity consumption fell by a third due to the Wuhan lockdown. It was only during March that consumption could return to the before-COVID level. Demand and supply shock following the lockdown forced to companies shut down their operations [5], resulting in a decrease in corporate financial performance [3,4,6,7,8]. Some industries, such as manufacturing, tourism, hospitality, transportation, film, and entertainment, were more severely affected, while other sectors, such as technology and e-commerce, enjoyed the decrease in physical interaction [1,4,9,10,11].

Studies on organizational slack, which represents corporate resources in excess of what is required for efficient operation [12,13,14], have suggested that slack provides a buffer against environmental fluctuations [15]. Furthermore, as slack resource absorbs external volatility, a company can protect its technological core [16,17]. Slack resource allows innovation that is not permitted when resources are scarce [12,14]. These effects—buffering, providing strategic flexibility, and protecting the core organization—reduce the impact of a recession [18]. One can expect a similar role of slack in the COVID-19 phase, which was confirmed in a study using a US sample [7].

Now that global trade and exchanges have significantly increased, similar crises will become more frequent and severe. If we cannot prevent them, attention should be paid to minimizing the damage. Therefore, we investigated slack as a mitigating device for the adverse effect of COVID-19 on businesses. Specifically, we used a Chinese sample where COVID-19 first developed as a pandemic and tested the buffering effect of financial slack during the middle periods of the COVID-19 pandemic. Unlike Shen et al. [4] and Li’s [7] study, which used a dummy variable for the period during which COVID-19 was first significantly affected, our study employed the quarterly number of patients in the province to which the company belongs and tested the impact of COVID-19 and slack on companies. We also looked at the differential effects of corporate financial health and industry-specific differences in the role of slack. Expanding on what early COVID-19 studies have focused on during the initial quarters of the outbreak, by testing the year 2021, we intended a follow-up study to look at the effects of our test variables when social adaptation to COVID-19 has occurred chronically.

Test results are as follows. First, consistent with the studies of the pandemic or natural disasters on firms [1,3,4,6,7], the COVID-19 outbreak suppressed the performance of Chinese listed companies during 2021. Consistent with the studies on slack resources [14,15,18,19,20], financial slack showed a positive relationship with corporate financial performance. Although not consistent across models, test results of intersection variables of COVID-19 and the slack resources in the full sample were positive.

Second, we discovered that this positive effect was more consistently observable in financially constrained firms. Because financially constrained firms face difficulties with external funding, slack as internal financing could help overcome the economic shock posed by the COVID-19 pandemic. Slack related to the external financing capacity also reduced the impact of the COVID-19 shock on financial performance.

Third, this intersection effect was only found in industries vulnerable to COVID-19. Studies have pointed out that the negative impact of COVID-19 is mainly on tourism, hospitality, restaurants, entertainment, transportation, aviation, energy and gas, and manufacturing industries [1,3,4,9,10,11,21]. Conversely, some industries, such as technology and e-commerce, were among the sectors that benefited from COVID-19 [1,9,11]. This study confirms that the negative effect of COVID-19 on financial performance and the buffering effect of slack resources only occurs in firms in these vulnerable industries. In all models, we included square terms to control for a possible non-linearity [14,19,20,22,23,24].

Existing slack resource studies have yet to pay much attention to the effects of slack resources under economic, environmental, and health shocks; particularly, studies linking the impact of COVID-19 are uncommon. Li (2021), one of the rare COVID-19 studies that fall within the slack resource literature, tested the initial outbreak using a U.S. sample. Since this paper is a study using data from China in the year following the outbreak, when and where COVID-19 has entered under relatively stable control, both the pandemic’s adverse effects and the mitigating effect of slack may become relatively small compared with other regions. Nevertheless, since this study found the effect of slack with a certain degree of clarity, we believe that it confirmed the robustness of the buffering effect of slack resources during the crises such as COVID-19. We also believe that this study contributes to the literature in that we employed a more precise model by using the number of COVID-19 patients instead of the quarter dummies only showing the difference in the averages between the initial outbreak period and the previous periods.

## 2. Literature Review and Hypothesis Development

### 2.1. Pandemic and Firm Performance

After the new coronavirus disease was first identified in December 2019, COVID-19 rapidly spread to the world, causing a vast economic breakdown and leading to the worst global recession since 1930 [3,4]. The infection controls such as social distancing and lockdowns seriously impacted world transportation and economic activities, bringing tremendous pressures on firms’ daily operations [3,4,6].

Manufacturing firms scaled down their production and decreased operations to minimal levels [1,6]. In total, 94% of the Fortune 1000 companies were affected by coronavirus-driven supply chain disruptions in February 2020 [25]. The supply availability and market demand in global supply chains were dramatically reduced in the early stage of the COVID-19 outbreak. COVID-19 has also significantly impacted China’s economy and businesses during 2020, as seen in Table 1.

The slowdown in growth caused by the restrictions on economic activities had a broader and more negative global impact on firms’ internal and external operations than any previous epidemics because of the disruption of the supply and demand chain [1,26]. Based on the market signal mechanism, COVID-19 impacted not only firms’ operations [1,3] but also their stock prices [27,28,29,30] and financial performances [4,7]. 

Studies on former pandemics also have discovered negative impacts due to blows, such as a decline in the workforce, restrictions on the movement of factors of production, and a decrease in supply and demand [31,32,33,34]. Past pandemics have also caused economic and stock market declines [35,36,37,38]. COVID-19 may be essentially indistinguishable from past pandemics, despite the difference in magnitude. Pandemics have a nationwide effect like a natural disaster [39]. Therefore, businesses are affected to a certain extent (See Appendix A for a table of summaries of some key and recent studies on the related subject).

### 2.2. Financial Slack and Firm Performance

Organizational slack is an excess resource that provides buffers or opportunities [13]. It refers to resources above the optimal level possessed by the company [12,13,14]. Slack consists of excess resources currently embedded within the firm and assimilated into the technical design of the organization, taking the form of excess cash, redundant employees, unused capacity, and unnecessary capital expenditures [12]. These resources indicate unexploited opportunities to increase output in the future [14].

According to Bourgeois [12], slack can bring some useful effects to an organization. For example, slack can retain employees by paying more than optimal wages. By allowing resources to be allocated to non-essential projects or loosening restrictions on expenditure, slack allows members within the organization to pursue different goals and thus avoids goal conflicts within the organization. It provides a buffer of resource input, reducing the need for tight coordination between organizations and activities [12].

Studies generally have explained the buffering function of slack in terms of mitigating the impact caused by changes in the environment surrounding companies. By smoothing or absorbing financial fluctuations, slack minimizes lack of funds and, as a result, maintains activities such as production, investment, and employment and protects the technical core of the organization [16,40]. Most importantly, slack drives innovation by allowing organizations to experiment with new strategies, and as a result, new products and markets can be pioneered [12,14]. It can facilitate creative behavior in the strategic change context [41].

These effects of slack have been confirmed as having a significant correlation with financial performance. Singh’s [15] study showed that performance determines slack and slack increases a firm’s risk-taking. Bromiley [22] tested the relationship between slack and risk-taking and discovered a U-shape nonlinear relationship. However, more studies showed an inversed U-shape relationship between slack and corporate outcomes such as innovation or financial performance [14,19,20,42] (See Appendix A for a table of summaries of some key and recent studies on the related subject).

### 2.3. Hypothesis Development

Regarding the role of organizational slack, studies have anticipated the positive impact of slack on an organization in terms of buffering and innovation [12,14,15,16,19,40]. Latham and Braun [18] suggested that slack can mitigate the negative impact of recession for three reasons: buffering, strategic flexibility, and acquiring the normality of an organization’s technical core. First, slack may function as a “rainy-day fund,” allowing a firm to resist uncertain environmental risks. Second, with slack, companies can more flexibly apply strategies to respond to economic downturns, such as maintaining or launching new resource investments to achieve a comparative advantage. Third, slack protects the technical core of an organization, such as investment and human resources, from environmental variability by maintaining the normality of corporate activities during a recession when external resources are depleted.

Currently, there are not many studies examining the effect of slack on firm performance in the COVID-19 environment. Li’s [7] US study focused on the outbreaks in early 2020 and showed that slack mitigated ROA reduction during the lockdown. Shen et al.’s [4] Chinese study employed a similar setting, but they did not test slack. As Chinese studies have also confirmed slack’s effect on firm performance [19,24,43,44], this study predicts that the impact of COVID-19 on Chinese companies will be less for companies with abundant slack.

**Hypothesis** **1.**
*Organizational slack mitigates the adverse effect of COVID-19 on corporate financial performance.*


According to Cheng et al. [45], financial constraints refer to market frictions that prevent companies from raising funds for all their desired investments, that is, a state in which external financing becomes difficult. It arises from “credit constraints or inability to borrow, inability to issue equity, the dependence of bank loan, or illiquidity of assets” [46]. Moreover, pandemics have caused economic declines, including a decrease in corporate activities and performances [3,4,7,31,32,37,38], which will raise the hurdles of corporate financing. Thus, it can be predicted that companies facing financial constraints will be more vulnerable to the pandemic, where slack can act as internal funding to protect its technical core. In other words, the more financially vulnerable you are, the stronger the buffering effect of slack against COVID-19 will be.

**Hypothesis** **2.**
*Organizational slack of financially constrained firms mitigates the adverse effect of COVID-19 on corporate financial performance.*


The impact of the pandemic may not have been the same across industries. In Hassan et al.’s [1] study using data from 82 countries, the demand decline was significant in the transportation, energy and utilities, and industrial goods and services sectors and small in the technology, healthcare, and education sectors. Gu et al.’s [3] Chinese study showed that significant reductions in electricity use had been found in the mining and quarrying, manufacturing, hospitality, and culture/sports/entertainment industries. A study by Shen et al. [4] on Chinese companies mentioned tourism, entertainment, catering retail, transportation, realty business, construction, accommodation, and export manufacturing industries have been negatively affected by the pandemic. 

Researchers such as Davis et al. [9], Mazur et al. [21], Al Guindy [11], and Baek and Lee [10] suggested industries were vulnerable to the pandemic, including transportation and aviation, hospitality, restaurant, entertainment, energy and/or petroleum industries. On the other hand, studies classified technology, e-commerce, web-based services, raw materials, and/or real estate industries as beneficiaries of the outbreak [1,9,11]. Reduced activity and work from home may cause decreased demand for transportation, travel, and other related service industries mentioned above. Therefore, we propose Hypothesis 3 as follows. In industries greatly affected by COVID-19, it can be expected that the difference between companies with margins from slacks and those that do not will be more clearly revealed.

**Hypothesis** **3.**
*Organizational slack in vulnerable industries to COVID-19 mitigates the adverse effect of COVID-19 on corporate financial performance.*


The structure in which research concepts are connected to form hypotheses is shown in (Figure 1) below.

## 3. Research Design

### 3.1. Sample Selection

Our data consists of 9306 Chinese public firm-quarters during 2021 from Accounting Research (CSMAR) databases. We selected the year 2021 because we intended a follow-up study to Li [7]. Both Shen et al.’s [4] study on Chinese companies and Li’s [7] study on US firms tested the pandemic’s effect on ROA around the early quarters of 2020, which was the period when the impact of the COVID-19 outbreak was the greatest. The problem is that this extreme effect was relatively temporary. Gu et al.’s [3] Chinese study suggested that businesses may have been recovering from the second and third quarters of 2020. The need for follow-up studies with data after 2021 is raised when the extreme moment had passed, but the pandemic still had an impact. Therefore, we selected the four quarters of 2021 as our study period.

### 3.2. Variable Measurements

#### 3.2.1. Dependent Variable

Many studies on the relationship between slack and firm performance have chosen ROA as a variable for performance [15,18,22,23,24,42,43,44,47]. Studies on the COVID-19 pandemic and firm performance also employed ROA as a sole dependent variable [4,7]. Since ROA is the most frequently selected performance variable in the related literature, we chose ROA as the corporate financial performance.

#### 3.2.2. Independent Variable

Our independent variable, COVID-19 (*covid*), is the quarterly cumulative number of COVID-19 patient cases in the region where the company is located. If the region is further subdivided, the variance is not constant, such as zero patients for a long period of time in one specific city and a number of patients in the other specific city. We decided to stabilize the variables by collecting them at the province level. This is an improvement compared with the related studies that had used dummy variables for the period when COVID-19 outbreaks occurred. Separating a specific period of COVID-19 from the rest was appropriate in early studies of the COVID-19 outbreak but not in follow-up studies when the COVID-19 persisted. The relevance of the period dummy variable may decrease as the period becomes longer. It becomes challenging to assume that the variable reflects only the impact of COVID-19 as the likelihood of the influence of other factors increases.

#### 3.2.3. Moderating Variable

This study examined how the relationship between COVID-19 and performance changes by introducing organizational slack as a moderating variable. Studies have generally classified slack into three categories [13,15,18,23,24,48,49]. The first group of slacks is referred to by the following terms: available slack, unabsorbed slack, or high-discretionary slack. It is an uncommitted slack or excess liquidity that has already been monetized or is ready to be put into business (e.g., cash and marketable securities). The second group means a slack in a state of being executed in an organization and is called recoverable slack, absorbed slack, or low-discretionary slack. It is a committed slack that has been assigned to use for a specific purpose, such as receivables, inventories, or excess operational cost. Finally, potential slack has the lowest degree of easiness of recovery or discretionary and generally means the room to finance additional external debt or capital. We selected the current ratio for available or unabsorbed slack, SG&A expenses ratio for unavailable or absorbed slack, and equity-to-debt ratio for the potential slack because they are the most frequently selected variables in the organizational slack-performance studies [15,18,22,23,24,43,50].

#### 3.2.4. Control Variable

We controlled the following company characteristic variables. Firm size (*size*), measured as the logarithm of common equity at the beginning of month t, reflects differences in financing options depending on company size [51]. The beta coefficient (*beta*) captures the price movements of individual stocks relative to the overall stock market. To control the growth potential, we selected the sales growth ratio (*grw*). We also included years from the establishment (*age*), as redundant enterprise resources increase with age, and the increase of redundant resources will lead to the enhancement of enterprise viability and performance [20]. To control the firm profitability environment, we selected a dummy variable for an unprofitable firm (*loss*). Finally, fixed effects related to year and industry were controlled by adding quarter and industry dummy. To control for extreme values, we applied 10% winsorization to all variables. Definitions of all variables are presented in Table 2.

### 3.3. Research Model

The test model is presented below. We predict negative values for *a*_1_ and positive values for *a*_2_. The research question of Hypothesis 1 is tested from the coefficient of *covid* × *slack*, and we expect positive values, implying that the more free resources are accessible, the less the company’s damage from COVID-19 is.
$$roa=\alpha_{0}+\alpha_{1}covid+\alpha_{2}slack+\alpha_{3}covid*slack+\alpha_{4}size+\alpha_{5}beta+\alpha_{6}grw\;+\alpha_{7}age+\alpha_{8}loss+\alpha_{9}Indus+\alpha_{10}Quarter+\varepsilon_{t}$$

For Hypotheses 2 and 3, we applied the above model separately to the two subsamples that were either financially constrained or not, and that belong to industries negatively influenced by COVID-19 or not. We classified firms as financially constrained based on Hadlock and Pierce [52]. Our definition of the industries negatively impacted by COVID-19 is based on the literature [1,3,4,9,10,11,21].

## 4. Test Results

Table 3 shows descriptive statistics for the variables. The means of the test sample are 0.024 for *roa*, 0.037 for *covid*, 1.997 for *cr*, 0.060 for *sga*, and 1.770 for *eq_debt*. The low standard deviation of ROA indicates that companies’ quarterly performance did not fluctuate significantly during the test period. The average logarithm of total assets (*size*) is 22.505; the average beta is almost 1. The average quarterly growth rate is 5.1%, indicating that Chinese companies in 2021 are relatively well-adapted to the pandemic.

Table 4 shows the correlation among the test variables. In the table, *roa* and *cr/eq_debt* are positively correlated, but *roa* and *sgar* are negatively correlated. This is consistent with a number of Chinese and Asian studies showing that unabsorbed slack showed a positive relationship with performance [19,23,24,43,44]., whereas absorbed slack had a negative relationship with performance [19,20,21,22,23,24]. Since the relationship between *roa* and *covid* is insignificant, Chinese companies in 2021 may not be affected by the incidence of patients if we rely solely on the correlation analysis. In addition, *cr* and *sgar* have positive relationships with *covid*, implying that businesses maintain internal capital to prepare for the COVID-19 crisis. It is not possible through correlation tests to prove the hypotheses that suggest causality, so it is necessary to perform regression analyses that include control for various company-related or other fixed characteristics.

Table 5 presents test results of the OLS regression analysis that show the moderating effects of slack on the relationships between COVID-19 and firm performance. Model 1 indicates that COVID-19 significantly reduced the performance of Chinese companies in 2021, just as in the US and China in early 2020 [4,7]. Models 2, 7, and 12 show that all three types of slack increase firm performance, consistent with the literature that proved the positive relationship between slack resources and firm performance [13,15,18,19,23,24]. In models 3, 8, and 13, COVID-19 and slack are tested together and provide consistent results. Models 4, 9, and 14 provide test results of the moderating effect of slack related to Hypothesis 1. Except for model 9, there were no significant results for the intersection variables, so from Table 5, Hypothesis 1 was only partially confirmed.

In models 5, 6, 10, 11, 15, and 16, we added square terms of the slack variables. Unlike organization theory, which assumes a positive relationship between slack and performance, agency theory predicts the opposite, a negative relationship. Slack may mean low efficiency of resource utilization, resulting from a manager’s agency problem [14,18,19]. Excessive slack can cause agency problems and lower performance [20], and too little slack does not allow experimentation for innovation [14], implying an inversed U-shape between slack and performance [14,19,20,23,24,42]. To control such nonlinearity, studies construct a model that additionally controls the square terms [20,22,23]. In the model with square terms, the coefficients of intersection variables are statistically significant and positive in Models 6 and 11, supporting Hypothesis 1.

Table 6 and Table 7 show the test results of the OLS regression for each of the financially distressed and non-distressed subsamples. Control variables are omitted from Table 6 due to space limitation. The intersection variables in the financially distressed subsample are positively significant for all three slack alternatives (Table 6 models 4, 6, 9, 11, 14, and 16). However, for the non-constraint companies, the intersection is significant only for *sga* (Table 7 models 4, 6, 9, 11, 14, and 16). The more limited a firm’s financing environment, the greater the mitigation effect of slack on the negative impact of COVID-19, as slack means internal accessibility of excess resources. Test results of Table 6 and Table 7 support Hypothesis 2.

Table 8 and Table 9 show test results of H3 for industries negatively affected by COVID-19 and those not. We included all industries classified as negatively affected if they are mentioned in the literature. Manufacturing industries were basically included as affected except for those mentioned as beneficiary industries, such as IT and e-commerce. Thus, the sample size for affected industries is greater than that of non-affected industries. Firm-quarters in the negatively affected industries presented in Table 8 comprise 7893 samples, while non-damaged ones in Table 9 have a smaller firm-quarters of 1413. In Table 9, no significant results were observable for *covid* in all models, whereas in Table 8, all *covid* are statistically significant.

Test results show that the effect in the full sample is due to the negatively affected industry. In Table 8, the coefficient values of intersection are significant in Models 9 and 11, which is similar to the results in Table 5. However, the intersections in Table 9 are not significant in all models (models 4, 6, 9, 11, 14, and 16).

## 5. Discussion and Conclusions

### 5.1. Findings of This Study

The COVID-19 pandemic has caused nationwide lockdowns and overall activity decline, creating an unfavorable situation for enterprises. When facing the rapidly changing environment and having intense pressures from supply and demand shortages, firms need to take action in response to the outbreak. However, their responses are contingent upon the availability of internal resources, such as excess cash and underutilized capacity [53].

During the COVID-19 outbreak, the slack may allow firms to react swiftly to external pressures and seize emerging opportunities. For example, as the demand for personal protective equipment increases, some firms make a quick transition to produce masks or other COVID-19-related products. Firms having extra human resources, production capacity, and underutilized equipment are better positioned to quickly analyze their operations and train their workforce. 

We tested whether organizational slack offsets the negative impact of COVID-19 on corporate performance. Test results did not have the expected effects consistently. The inconsistency across the full sample may be because we selected companies of 2021 when COVID-19 is controlled relatively constantly. However, as we apply the ideas of Hypotheses 2 and 3, results present a more apparent moderating effect for the specific subsamples. This study found a certain moderating effect of slack under the influence of COVID-19, and in particular, this effect was more evident in the subsample with financing and industrial weakness.

### 5.2. Implications to the Management, Policymakers, and Society

We have witnessed through COVID-19 that public health problems can escalate into significant economic crises at a global level, and it is predicted that such crises will be repeated in the future. Moreover, the frequency of significant environmental impacts due to climate change will become more frequent. Generally speaking, in the past many decades, economic or social volatility has repeatedly shown. Because of the acceleration of the international economic and physical connections between countries and regions, now the possibility of economic, environmental, and health-related problems occurring in one region and spreading to other areas has become higher than before. 

Therefore, the need to prepare for such impacts on businesses, politics, and communities is increasing. Through the case of COVID-19, this study confirmed that economic participants are more resistant to external uncertainties if they maintain sufficient internal financial buffers. This finding suggests to managers the managerial necessity of maintaining internal free resources at a level enough to prepare for repeated uncertainties, to policymakers the policy necessity in specific industries to ensure that such a buffer is sufficiently maintained, and to society the need to maintain some portion of resources at stable, low-risk, high-accessibility points.

### 5.3. Contributions and Limitations

This paper is a study confirming the effect of slack in a situation when the pressure of a pandemic has become a daily thing. The contributions of this study to the literature are as follows. First, existing studies on slack resources generally focus on normal conditions, and research under economic, environmental, or health impacts and uncertainties is relatively less common. This study has contributed to the literature in that it is one of the few studies that tested slack’s moderating effect on the relationship between COVID-19 and performance. Second, Li’s [7] US study tested the initial impact of the outbreak only. The long and consistent impact of COVID-19, extended beyond the time of initial shock, creates the need for follow-up studies. COVID-19 management in some Asian countries, including China, has been more successful and stable than in many other regions. Therefore, the effects surrounding COVID-19 may be reduced in these samples, so it is worth doing a separate study. We were able to confirm the robust effect of slack in data from China for 2021. Third, many studies on pandemics use the period dummy, which simplifies the assumption that only the selected event’s effect exists throughout the test period. This approach is helpful for events of short duration, but the dummy approach may encounter certain limitations for longer-term crises. As COVID-19 lasts beyond a year or two, it becomes less accurate to dummy the test period and see the difference between the two periods’ averages. This study accumulated patient data and tested the slope of the linear relationship between the variables in question.

The limitation of this study is that we did not observe longer-term effects because we initiated the study by collecting a one-year sample shortly after the fourth quarter of 2021. In addition, in our empirical models, the effects of proxies did not appear consistently in all models. That is, test results can be affected by what the researcher chooses as the slack variable. Therefore, further consideration of variables and longer study periods are requested in subsequent studies.

## Figures and Tables

**Figure 1 ijerph-19-14354-f001:**
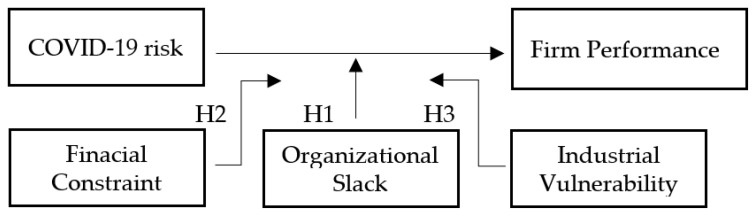
Conceptual structure.

**Table 1 ijerph-19-14354-t001:** China’s quarterly GDP growth during 2020.

Quarter	GDP	GDP Growth
March 2020	205,727	−6.8
June 2020	248,985.1	3.2
September 2020	264,976.3	4.9
December 2020	296,297.8	6.5

**Table 2 ijerph-19-14354-t002:** Variable definitions.

Symbol		Variable Name	Definition
*roa*		Return on assets	Net income divided by total assets
*covid*		COVID-19 risk	Natural logarithm of COVID-19 infection number
*Slack*	*cr*	Current ratio	Current assets divided by current liability
	*sga*	SG&A ratio	Selling, general and administrative expenses divided by sales
	*eq_debt*	Equity-to-debt ratio	Net asset divided by total liabilities
*sga*		Financially constraint firm dummy	A dummy variable assigning 1 for the financially constraint companies and 0 otherwise, based on Hadlock and Pierce [52]’s constraint model, the model is saa = 0.737 × size + 0.043 × size^2^ − 0.040 × age
*indd*		Vulnerable Industries dummy	A dummy variable assigning 1 for industries negatively affected by COVID-19 and 0 otherwise
*size*		Firm size	Natural logarithm of total asset
*beta*		Firm risk	Beta coefficient
*grw*		Sales growth	One quarter percentage growth in sales
*loss*		Net loss dummy	A dummy variable assigning 1 for firms reporting net losses and 0 otherwise
*age*		Firm age	Natural logarithm of the years from the corporate establishment
*indus*		Industry effect	Industry dummies
*quarter*		Quarter effect	Quarter dummies

**Table 3 ijerph-19-14354-t003:** Descriptive Statistics.

Variables	N	Mean	Median	Min	Max	Standard Deviation
*roa*	9306	0.024	0.018	−0.009	0.067	0.024
*covid*	9306	0.037	0.037	0.016	0.060	0.013
*cr*	9306	1.977	1.624	0.831	4.184	1.062
*sga*	9306	0.060	0.033	0.006	0.196	0.061
*eq_debt*	9306	1.770	1.273	0.418	4.666	1.350
*saa*	9306	0.500	0.500	0	1	0.497
*indd*	9306	0.848	1	0	1	0.359
*size*	9306	22.505	22.363	21.015	24.399	1.084
*beta*	9306	0.988	0.970	0.596	1.466	0.278
*grw*	9306	0.051	0.025	−0.440	0.676	0.300
*loss*	9306	0.137	0	0	1	0.343
*age*	9306	3.077	3.091	2.708	3.367	0.215

*roa*: return on assets; *covid*: natural logarithm of COVID-19 infection number; *cr*: current ratio, current assets divided by current liability; *sga*: SG&A ratio, SG&A expenses divided by sales; *eq_debt*: equity-to-debt ratio, net asset divided by total liabilities; *saa*: a dummy variable assigning 1 for the financially constrained companies and 0 otherwise, based on Hadlock and Pierce’s [52] constraint model, the model is *saa* = 0.737 × *size* + 0.043 × *size*^2^ − 0.040 × *age*; *indd*: a dummy variable assigning 1 for industries negatively affected by COVID-19 and 0 otherwise; *size*: natural logarithm of total asset; *beta*: beta coefficient; *grw*: growth in sales; *loss*: a dummy variable assigning 1 for firms reporting net losses and 0 otherwise; *age*: natural logarithm of the years from the corporate establishment.

**Table 4 ijerph-19-14354-t004:** Correlation Matrix.

	*roa*	*covid*	*cr*	*sgar*	*eq_debt*	*indd*	*saa*	*size*	*beta*	*grw*	*loss*	*age*
*roa*	1											

*covid*	0.001	1										
	0.928											
*cr*	0.214	0.022	1									
	<0.0001	0.036										
*sgar*	−0.032	0.022	0.209	1								
	0.002	0.034	<0.0001									
*eq_debt*	0.244	0.000	0.859	0.221	1							
	<0.0001	0.998	<0.0001	<0.0001								
*indd*	−0.014	−0.004	−0.189	−0.395	−0.214	1						
	0.191	0.732	<0.0001	<0.0001	<0.0001							
*saa*	0.095	−0.047	−0.284	−0.095	−0.281	−0.004	1					
	<0.0001	<0.0001	<0.0001	<0.0001	<0.0001	0.704						
*size*	0.058	−0.048	−0.382	−0.184	−0.409	0.142	0.778	1				
	<0.0001	<0.0001	<0.0001	<0.0001	<0.0001	<0.0001	<0.0001					
*beta*	−0.018	0.024	−0.047	−0.012	−0.035	0.005	0.060	0.066	1			
	0.083	0.018	<0.0001	0.247	0.001	0.639	<0.0001	<0.0001				
*grw*	0.149	−0.043	−0.004	−0.057	0.000	−0.007	0.008	0.003	0.000	1		
	<0.0001	<0.0001	0.733	<0.0001	0.971	0.523	0.442	0.803	0.997			
*loss*	−0.518	0.011	−0.102	0.130	−0.100	−0.032	−0.114	−0.130	0.028	−0.083	1	
	<0.0001	0.273	<0.0001	<0.0001	<0.0001	0.002	<0.0001	<0.0001	0.006	<0.0001		
*age*	−0.064	0.002	−0.074	−0.025	−0.079	0.052	0.053	0.141	0.081	−0.011	0.017	1
	<0.0001	0.866	<0.0001	0.017	<0.0001	<0.0001	<0.0001	<0.0001	<0.0001	0.311	0.094	

**Table 5 ijerph-19-14354-t005:** Regression results for H1.

**Panel A**	**(1)** ** *roa* **	**(2)** ** *roa* **	**(3)** ** *roa* **	**(4)** ** *roa* **	**(5)** ** *roa* **	**(6)** ** *roa* **
*covid*	−0.037 **		−0.038 ***	−0.071 **		−0.087 ***
	(−2.562)		(−2.651)	(−2.414)		(−2.976)
*cr*		0.004 ***	0.004 ***	0.004 ***	0.013 ***	0.012 ***
		(22.395)	(22.404)	(7.010)	(13.740)	(11.871)
*covid*cr*				0.017		0.022 *
				(1.293)		(1.691)
*cr^2^*					−0.002 ***	−0.002 ***
					(−9.460)	(-9.656)
*size*	0.000 **	0.002 ***	0.002 ***	0.002 ***	0.002 ***	0.002 ***
	(2.119)	(10.577)	(10.450)	(10.463)	(11.573)	(11.469)
*beta*	0.000	0.000	0.000	0.000	0.000	0.000
	(0.110)	(0.526)	(0.601)	(0.580)	(0.468)	(0.528)
*grw*	0.000	0.000	0.000	0.000	0.000	0.000
	(0.075)	(0.473)	(0.441)	(0.443)	(0.474)	(0.438)
*loss*	−0.034 ***	−0.032 ***	−0.032 ***	−0.032 ***	−0.031 ***	−0.031 ***
	(−61.540)	(−58.488)	(−58.513)	(−58.499)	(−57.182)	(−57.172)
*age*	−0.004 ***	−0.004 ***	−0.004 ***	−0.004 ***	−0.004 ***	-0.004 ***
	(−4.793)	(−4.777)	(−4.760)	(−4.744)	(−4.403)	(−4.355)
Constant	0.019 ***	−0.029 ***	−0.027 ***	−0.026 ***	−0.043 ***	−0.040 ***
	(4.113)	(−5.896)	(−5.478)	(−5.175)	(−8.436)	(−7.660)
Ind. & Qrt.	controlled	controlled	controlled	controlled	controlled	controlled
Observations	9306	9306	9306	9306	9306	9306
Adjusted R^2^	0.440	0.468	0.468	0.468	0.473	0.474
F	522.4	585.5	547.3	513.3	557.7	493.3
**Panel B**	**(7)** ** *roa* **	**(8)** ** *roa* **	**(9)** ** *roa* **	**(10)** ** *roa* **	**(11)** ** *roa* **	
*covid*		−0.038 ***	−0.082 ***		−0.079 ***	
		(−2.621)	(−4.119)		(−3.951)	
*sga*	0.023 ***	0.023 ***	−0.005	−0.049 ***	−0.074 ***	
	(7.316)	(7.337)	(−0.495)	(−3.971)	(−4.976)	
*covid*sga*			0.752 ***		0.718 ***	
			(3.220)		(3.078)	
*sga^2^*				0.368 ***	0.360 ***	
				(6.045)	(5.928)	
*size*	0.001 ***	0.001 ***	0.001 ***	0.000 ***	0.000 **	
	(3.374)	(3.247)	(3.282)	(2.646)	(2.572)	
*beta*	0.000	0.000	0.000	0.000	0.000	
	(0.154)	(0.228)	(0.183)	(0.043)	(0.074)	
*grw*	0.000	0.000	0.000	0.000	0.000	
	(0.433)	(0.402)	(0.358)	(0.384)	(0.312)	
*loss*	−0.034 ***	−0.034 ***	−0.034 ***	−0.034 ***	−0.034 ***	
	(−62.096)	(−62.123)	(−62.199)	(−62.112)	(−62.210)	
*age*	−0.004 ***	−0.004 ***	−0.004 ***	−0.004 ***	−0.004 ***	
	(−4.739)	(−4.723)	(−4.641)	(−4.972)	(−4.873)	
Constant	0.010 **	0.012 **	0.013 ***	0.016 ***	0.019 ***	
	(2.194)	(2.556)	(2.822)	(3.321)	(3.877)	
Ind. & Qrt.	controlled	controlled	controlled	controlled	controlled	
Observations	9306	9306	9306	9306	9306	
Adjusted R^2^	0.442	0.443	0.443	0.445	0.445	
F	528.4	493.9	464.2	497.5	440.5	
**Panel C**	**(12)** ** *roa* **	**(13)** ** *roa* **	**(14)** ** *roa* **	**(15)** ** *roa* **	**(16)** ** *roa* **	
*covid*		−0.029 **	−0.034		−0.033	
		(−2.063)	(−1.470)		(−1.489)	
*eq_debt*	0.004 ***	0.004 ***	0.004 ***	0.015 ***	0.015 ***	
	(26.581)	(26.533)	(9.539)	(24.323)	(20.624)	
*covid*eq_debt*			0.003		0.003	
			(0.254)		(0.262)	
*eq_debt^2^*				−0.002 ***	−0.002 ***	
				(−18.330)	(−18.331)	
*size*	0.002 ***	0.002 ***	0.002 ***	0.003 ***	0.003 ***	
	(12.680)	(12.552)	(12.553)	(16.065)	(15.934)	
*beta*	0.000	0.000	0.000	0.000	0.000	
	(0.271)	(0.329)	(0.324)	(0.120)	(0.174)	
*grw*	0.000	0.000	0.000	0.001	0.001	
	(0.462)	(0.436)	(0.434)	(0.833)	(0.804)	
*loss*	−0.031 ***	−0.031 ***	−0.031 ***	−0.030 ***	−0.030 ***	
	(−58.371)	(−58.393)	(−58.387)	(−56.552)	(−56.569)	
*age*	−0.004 ***	−0.004 ***	−0.004 ***	−0.004 ***	−0.004 ***	
	(−4.933)	(−4.919)	(−4.914)	(−4.302)	(−4.283)	
Constant	−0.036 ***	−0.035 ***	−0.035 ***	−0.061 ***	−0.060 ***	
	(−7.500)	(−7.117)	(−7.030)	(−12.406)	(−11.870)	
Ind. & Qrt.	controlled	controlled	controlled	controlled	controlled	
Observations	9306	9306	9306	9306	9306	
Adjusted R^2^	0.479	0.479	0.479	0.497	0.497	
F	611.7	571.4	535.6	613.9	542.1	

*, ** and *** denote significance at *p* < 0.1, <0.05 and <0.01, respectively, and *t*-values in parentheses. Ind. means *indus* and Qrt. means *quarter*. Refer to Table 2 for other variable definitions.

**Table 6 ijerph-19-14354-t006:** Regression Results for H2: Financially constrained subsample.

**Panel A**	**(1)** ** *roa* **	**(2)** ** *roa* **	**(3)** ** *roa* **	**(4)** ** *roa* **	**(5)** ** *roa* **	**(6)** ** *roa* **
*covid*	−0.032		−0.028	−0.184 ***		−0.182 ***
	(−1.438)		(−1.297)	(−4.028)		(−3.999)
*cr*		0.006 ***	0.006 ***	0.003 ***	0.014 ***	0.010 ***
		(18.031)	(18.017)	(3.097)	(9.125)	(5.919)
*covid*cr*				0.096 ***		0.091 ***
				(3.866)		(3.664)
*cr^2^*					−0.002 ***	−0.002 ***
					(−5.096)	(−5.039)
Constant	0.055 ***	−0.011	−0.010	−0.004	−0.024 **	−0.018 *
	(5.492)	(−1.027)	(−0.946)	(−0.434)	(−2.269)	(−1.677)
Control var.	controlled	controlled	controlled	controlled	controlled	controlled
Ind. & Qrt.	controlled	controlled	controlled	controlled	controlled	controlled
Observations	4653	4653	4653	4653	4653	4653
Adjusted R^2^	0.424	0.466	0.466	0.468	0.469	0.471
F	218.5	258.7	241.6	228.2	244.6	217.5
**Panel B**	**(7)** ** *roa* **	**(8)** ** *roa* **	**(9)** ** *roa* **	**(10)** ** *roa* **	**(11)** ** *roa* **	
*covid*		−0.028	−0.078 ***		−0.077 ***	
		(−1.303)	(−2.679)		(−2.645)	
*sga*	0.042 ***	0.041 ***	0.008	0.000	−0.034	
	(8.334)	(8.311)	(0.560)	(0.015)	(−1.474)	
*covid*sga*			0.947 **		0.969 ***	
			(2.563)		(2.622)	
*sga^2^*				0.214 **	0.216 **	
				(2.246)	(2.260)	
Constant	0.035 ***	0.036 ***	0.038 ***	0.039 ***	0.041 ***	
	(3.508)	(3.584)	(3.730)	(3.803)	(4.022)	
Control var.	controlled	controlled	controlled	controlled	controlled	
Ind. & Qrt.	controlled	controlled	controlled	controlled	controlled	
Observations	4653	4653	4653	4653	4653	
Adjusted R^2^	0.433	0.433	0.434	0.434	0.435	
F	226.9	211.9	199.4	212.3	188.1	
**Panel C**	**(12)** ** *roa* **	**(13)** ** *roa* **	**(14)** ** *roa* **	**(15)** ** *roa* **	**(16)** ** *roa* **	
*covid*		−0.011	−0.091 ***		−0.076 **	
		(−0.554)	(−2.768)		(−2.380)	
*eq_debt*	0.006 ***	0.006 ***	0.004 ***	0.021 ***	0.019 ***	
	(23.601)	(23.556)	(5.879)	(22.183)	(16.490)	
*covid*eq_debt*			0.060 ***		0.045 **	
			(3.121)		(2.412)	
*eq_debt^2^*				−0.003 ***	−0.003 ***	
				(−16.143)	(−16.022)	
Constant	−0.032 ***	−0.032 ***	−0.029 ***	−0.077 ***	−0.074 ***	
	(−3.218)	(−3.170)	(−2.896)	(−7.570)	(−7.240)	
Control var.	controlled	controlled	controlled	controlled	controlled	
Ind. & Qrt.	controlled	controlled	controlled	controlled	controlled	
Observations	4653	4653	4653	4653	4653	
Adjusted R^2^	0.492	0.492	0.493	0.522	0.523	
F	287.5	268.3	252.7	302.6	267.7	

*, ** and *** denote significance at *p* < 0.1, <0.05 and <0.01, respectively, and *t*-values in parentheses. Refer to Table 2 and Table 5 for variable definitions.

**Table 7 ijerph-19-14354-t007:** Regression Results for H2: Financially unconstrained subsample.

**Panel A**	**(1)** ** *roa* **	**(2)** ** *roa* **	**(3)** ** *roa* **	**(4)** ** *roa* **	**(5)** ** *roa* **	**(6)** ** *roa* **
*covid*	−0.040 **		−0.042 **	−0.049		−0.065
	(−2.080)		(−2.243)	(−1.209)		(−1.609)
*cr*		0.003 ***	0.003 ***	0.003 ***	0.010 ***	0.010 ***
		(15.390)	(15.412)	(5.153)	(8.010)	(7.283)
*covid*cr*				0.003		0.008
				(0.191)		(0.510)
*cr^2^*					−0.001 ***	−0.001 ***
					(−5.366)	(−5.499)
Constant	0.036 ***	−0.005	−0.003	−0.003	−0.015	−0.011
	(3.865)	(−0.588)	(−0.326)	(−0.292)	(−1.542)	(−1.184)
Control var.	controlled	controlled	controlled	controlled	controlled	controlled
Ind. & Qrt.	controlled	controlled	controlled	controlled	controlled	controlled
Observations	4653	4653	4653	4653	4653	5167
Adjusted R^2^	0.453	0.476	0.477	0.476	0.479	0.479
F	306.0	336.4	314.6	294.9	317.6	280.9
**Panel B**	**(7)** ** *roa* **	**(8)** ** *roa* **	**(9)** ** *roa* **	**(10)** ** *roa* **	**(11)** ** *roa* **	
*covid*		−0.041 **	−0.093 ***		−0.087 ***	
		(−2.142)	(−3.389)		(−3.207)	
*sga*	0.009 **	0.009 **	−0.021 *	−0.085 ***	−0.111 ***	
	(2.150)	(2.211)	(−1.755)	(−5.298)	(−5.727)	
*covid*sga*			0.803 ***		0.726 **	
			(2.648)		(2.402)	
*sga^2^*				0.476 ***	0.467 ***	
				(6.030)	(5.921)	
Constant	0.030 ***	0.032 ***	0.035 ***	0.039 ***	0.043 ***	
	(3.231)	(3.457)	(3.670)	(4.159)	(4.549)	
Control var.	controlled	controlled	controlled	controlled	controlled	
Ind. & Qrt.	controlled	controlled	controlled	controlled	controlled	
Observations	4653	4653	4653	4653	4653	
Adjusted R^2^	0.453	0.453	0.454	0.456	0.457	
F	306.1	286.2	269.0	290.0	256.9	
**Panel C**	**(12)** ** *roa* **	**(13)** ** *roa* **	**(14)** ** *roa* **	**(15)** ** *roa* **	**(16)** ** *roa* **	
*covid*		−0.037 *	−0.033		−0.032	
		(−1.946)	(−1.001)		(−0.969)	
*eq_debt*	0.003 ***	0.003 ***	0.003 ***	0.009 ***	0.009 ***	
	(16.676)	(16.657)	(5.907)	(11.492)	(9.957)	
*covid*eq_debt*			−0.002		−0.003	
			(−0.137)		(−0.201)	
*eq_debt^2^*				−0.001 ***	−0.001 ***	
				(−8.007)	(−8.015)	
Constant	−0.011	−0.009	−0.009	−0.023 **	−0.021 **	
	(−1.184)	(−0.943)	(−0.952)	(−2.461)	(−2.220)	
Control var.	controlled	controlled	controlled	controlled	controlled	
Ind. & Qrt.	controlled	controlled	controlled	controlled	controlled	
Observations	4653	4653	4653	4653	4653	
Adjusted R^2^	0.480	0.480	0.480	0.486	0.487	
F	341.8	319.4	299.4	327.2	289.1	

*, ** and *** denote significance at *p* < 0.1, <0.05 and <0.01, respectively, and *t*-values in parentheses. Refer to Table 2 and Table 5 for variable definitions.

**Table 8 ijerph-19-14354-t008:** Regression Results for H3: Industries impacted negatively by COVID-19.

**Panel A**	**(1)** ** *roa* **	**(2)** ** *roa* **	**(3)** ** *roa* **	**(4)** ** *roa* **	**(5)** ** *roa* **	**(6)** ** *roa* **
*covid*	−0.045 ***		−0.048 ***	−0.083 ***		−0.098 ***
	(−2.864)		(−3.146)	(−2.645)		(−3.131)
*cr*		0.004 ***	0.004 ***	0.004 ***	0.012 ***	0.011 ***
		(20.669)	(20.711)	(6.496)	(11.578)	(9.949)
*covid*cr*				0.019		0.023
				(1.271)		(1.586)
*cr^2^*					−0.001 ***	−0.002 ***
					(−7.470)	(−7.723)
Constant	0.015 **	−0.035 ***	−0.033 ***	−0.032 ***	−0.046 ***	−0.042 ***
	(2.533)	(−5.713)	(−5.319)	(−5.075)	(−7.286)	(−6.625)
Control var.	controlled	controlled	controlled	controlled	controlled	controlled
Ind. & Qrt.	controlled	controlled	controlled	controlled	controlled	controlled
Observations	7893	7893	7893	7893	7893	7893
Adjusted R^2^	0.434	0.463	0.463	0.463	0.466	0.467
F	433.2	486.1	454.8	426.6	460.6	408.0
**Panel B**	**(7)** ** *roa* **	**(8)** ** *roa* **	**(9)** ** *roa* **	**(10)** ** *roa* **	**(11)** ** *roa* **	
*covid*		−0.047 ***	−0.089 ***		−0.083 ***	
		(−2.971)	(−4.177)		(−3.899)	
*sga*	0.015 ***	0.015 ***	−0.017	−0.069 ***	−0.095 ***	
	(3.665)	(3.749)	(−1.476)	(−5.076)	(−5.584)	
*covid*sga*			0.859 ***		0.766 ***	
			(2.939)		(2.623)	
*sga^2^*				0.454 ***	0.441 ***	
				(6.430)	(6.249)	
Constant	0.008	0.010 *	0.012 *	0.016 ***	0.019 ***	
	(1.377)	(1.721)	(1.946)	(2.588)	(3.072)	
Control var.	controlled	controlled	controlled	controlled	controlled	
Ind. & Qrt.	controlled	controlled	controlled	controlled	controlled	
Observations	7893	7893	7893	7893	7893	
Adjusted R^2^	0.434	0.435	0.435	0.437	0.438	
F	433.8	405.9	381.4	409.7	363.0	
**Panel C**	**(12)** ** *roa* **	**(13)** ** *roa* **	**(14)** ** *roa* **	**(15)** ** *roa* **	**(16)** ** *roa* **	
*covid*		−0.042 ***	−0.049 **		−0.051 **	
		(−2.756)	(−2.039)		(−2.127)	
*eq_debt*	0.004 ***	0.004 ***	0.004 ***	0.015 ***	0.015 ***	
	(25.264)	(25.249)	(9.056)	(22.626)	(19.029)	
*covid*eq_debt*			0.005		0.004	
			(0.403)		(0.376)	
*eq_debt^2^*				−0.002 ***	−0.002 ***	
				(−16.733)	(−16.761)	
Constant	−0.046 ***	−0.044 ***	−0.044 ***	−0.073 ***	−0.071 ***	
	(−7.640)	(−7.254)	(−7.172)	(−11.839)	(−11.332)	
Control var.	controlled	controlled	controlled	controlled	controlled	
Ind. & Qrt.	controlled	controlled	controlled	controlled	controlled	
Observations	7893	7893	7893	7893	7893	
Adjusted R^2^	0.476	0.476	0.476	0.494	0.494	
F	512.7	479.5	449.5	514.2	454.6	

*, ** and *** denote significance at *p* < 0.1, <0.05 and <0.01, respectively, and *t*-values in parentheses. Refer to Table 2 and Table 5 for variable definitions.

**Table 9 ijerph-19-14354-t009:** Regression Results for H3: Industries not impacted negatively by COVID-19.

**Panel A**	**(1)** ** *roa* **	**(2)** ** *roa* **	**(3)** ** *roa* **	**(4)** ** *roa* **	**(5)** ** *roa* **	**(6)** ** *roa* **
*covid*	0.019		0.025	0.022		0.029
	(0.510)		(0.659)	(0.253)		(0.329)
*cr*		0.003 ***	0.003 ***	0.003 **	0.015 ***	0.015 ***
		(6.995)	(7.005)	(2.294)	(5.813)	(5.273)
*covid*cr*				0.001		0.001
				(0.031)		(0.016)
*cr^2^*					−0.002 ***	−0.002 ***
					(−4.720)	(−4.739)
Constant	−0.034 **	−0.058 ***	−0.059 ***	−0.059 ***	−0.073 ***	−0.074 ***
	(−2.556)	(−4.267)	(−4.316)	(−4.243)	(−5.281)	(−5.258)
Control var.	controlled	controlled	controlled	controlled	controlled	controlled
Ind. & Qrt.	controlled	controlled	controlled	controlled	controlled	controlled
Observations	1413	1413	1413	1413	1413	1413
Adjusted R^2^	0.490	0.507	0.506	0.506	0.514	0.514
F	136.4	146.0	132.7	121.6	136.8	115.7
**Panel B**	**(7)** ** *roa* **	**(8)** ** *roa* **	**(9)** ** *roa* **	**(10)** ** *roa* **	**(11)** ** *roa* **	
*covid*		0.020	−0.042		−0.038	
		(0.542)	(−0.587)		(−0.533)	
*sga*	0.027 ***	0.027 ***	0.007	0.064 *	0.045	
	(3.841)	(3.844)	(0.358)	(1.861)	(1.116)	
*covid*sga*			0.530		0.506	
			(1.030)		(0.982)	
*sga^2^*				−0.171	−0.167	
				(−1.112)	(−1.082)	
Constant	−0.035 ***	−0.036 ***	−0.034 **	−0.036 ***	−0.035 ***	
	(−2.645)	(−2.693)	(−2.521)	(−2.706)	(−2.588)	
Control var.	controlled	controlled	controlled	controlled	controlled	
Ind. & Qrt.	controlled	controlled	controlled	controlled	controlled	
Observations	1413	1413	1413	1413	1413	
Adjusted R^2^	0.495	0.494	0.494	0.495	0.495	
F	139.3	126.6	116.1	126.7	107.3	
**Panel C**	**(12)** ** *roa* **	**(13)** ** *roa* **	**(14)** ** *roa* **	**(15)** ** *roa* **	**(16)** ** *roa* **	
*covid*		0.036	0.075		0.098	
		(0.967)	(1.052)		(1.388)	
*eq_debt*	0.002 ***	0.002 ***	0.003 ***	0.011 ***	0.012 ***	
	(7.385)	(7.430)	(3.053)	(7.415)	(6.762)	
*covid*eq_debt*			−0.016		−0.021	
			(−0.643)		(−0.829)	
*eq_debt^2^*				−0.002 ***	−0.002 ***	
				(−5.968)	(−6.053)	
Constant	−0.059 ***	−0.061 ***	−0.062 ***	−0.070 ***	−0.074 ***	
	(−4.362)	(−4.455)	(−4.499)	(−5.241)	(−5.440)	
Control var.	controlled	controlled	controlled	controlled	controlled	
Ind. & Qrt.	controlled	controlled	controlled	controlled	controlled	
Observations	1413	1413	1413	1413	1413	
Adjusted R^2^	0.509	0.509	0.508	0.520	0.521	
F	147.1	133.8	122.6	140.3	118.9	

*, ** and *** denote significance at *p* < 0.1, <0.05 and <0.01, respectively, and *t*-values in parentheses. Refer to Table 2 and Table 5 for variable definitions.

## Data Availability

Not applicable.

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
