# Peer review of "Slack Resources, Corporate Performance, and COVID-19 Pandemic: Evidence from China"

_ijerph, 2022, doi:10.3390/ijerph192114354_

Round 1

Reviewer 1 Report

3.2 Variable Measurement

This part of the study should be improved. 

DEPENDENT VARIABLE. You choose ROA as a dependent variable of your study, but you should clarify the calculation method and the source of this variable (have you calculated the ROA from the corporate financial statement or have you used a database?).

INDIPENDENT VARIABLE. You choose COVID-19 cases in the region in wich the company are located. For the same reason of the dependent variable you should describe the source and the techniques that you have done for normalizing the firm-observations. 

MODERATING VARIABLE. You should clarify the formula for the determination of the variable and the source choosen. 

CONTROL VARIABLES. The same of the previous variable.

5. Conclusion and discussion 

The result of your study is very powerful and have strong research implication, but you should better underline the main limitations of your research and the future research paths. In the following statement you should clarify better in wich way you have contribute to improve the research methodology.

"As a result, we believe that this study has contributed 392 to the COVID-19 and organizational slack literature in two aspects: expanding the re- 393 search period and improving the research methodology"

Author Response

The answer is attached as a word file.

Reviewer 2 Report

Dear Authors

This is a well written and executed paper on a topical research area. However, it is recommended that the authors address the following:

 Please add the research objective in the Abstract. I would make the Abstract sharper: What is the problem, what have you done, what is the contribution? Contributions in fact are missing in the whole paper.

 The authors used very old citations, and it is suggested to read the recent papers and add citations. 

 The motivation for this study in the Introduction is not strong enough. Be bolder about why we should care about this and what the gap is in the literature.

 Please illustrate what changes to practise should be made as a result of this research and add to the body of knowledge.

 Please add a table summarising the previous highly cited studies, such as motivation, keywords, theories, methodology and analysis.

 When you use the abbreviation for the first time, please add the complete words. It is quite difficult to understand the text if you don't explain the abbreviations

 Please add a table of definitions for all the constructs used on the model

 Methodology section: The method and methodology employed should be explained further.  I am not very convinced why you have collected data from China, please add more justification.

Please add a reference to this sentence: “Organizational slack can bring several effects to an organization. Slack an organiza- 120 tion can retain employees by paying more than optimal wages. By allowing resources to 121 be allocated to non-essential projects or loosening restrictions on expenditure, slack allows 122 members within the organization to pursue different goals and thus avoids goal conflicts 123 within the organization”

The research question (what the author(s) try to achieve) and the new contribution of the study are missing. 

Hypotheses development is not strong. You are suggested to add conceptual model and formalize the causal mechanisms linking key constructs.

You require to convince the positioning of your study in the introductory section, identifying important gaps in the literature and considering contributions to existing theoretical knowledge, etc. 

Please build a section on managerial implications/policymakers implications/social implications.

For the theoretical and practical implications part: I would direct some questions to the authors:

- How has your view of your research topic changed?

-What is the ‘takeaway message” from this research?

- How do your findings relate to the literature in your field?

- What did you learn from doing this research? 

- What original contribution to knowledge do you feel that you have made?

 Again, in the managerial implication part, you should be able to answer to the question: what are the real implications?

Authors suggest superficial implications in the conclusion section. Please update the section. The research that authors have taken is very influential. However, the way they portrait is not explicit. No immediate challenges to overcome, no call to action has been given. 

   I wish the authors the very best in developing this stream of research.

Author Response

The answer is attached as a word file

Round 2

Reviewer 2 Report

Dear author(s),

I'm happy to see that you took into account my suggestions and that you strongly revised the paper. 

I think that it can be accepted for pubblication.

Best Regards